

# Measurement-induced entanglement phase transitions in variational quantum circuits

Roeland Wiersema[1,2*], Cunlu Zhou[1,3,4],
Juan Felipe Carrasquilla[1,2,5] and Yong Baek Kim[5]

**1** Vector Institute, MaRS Centre, Toronto, Ontario, M5G 1M1, Canada
**2** Department of Physics and Astronomy, University of Waterloo, Ontario, N2L 3G1, Canada
**3** Department of Physics and Astronomy and Center for Quantum Information and Control,
CQuIC, University of New Mexico, Albuquerque, New Mexico 87131, USA
**4** Department of Computer Science, University of Toronto, Ontario, M5T 3A1, Canada
**5** Department of Physics, University of Toronto, Ontario M5S 1A7, Canada

★ rwiersema@uwaterloo.ca

## Abstract

Variational quantum algorithms (VQAs), which classically optimize a parametrized quantum circuit to solve a computational task, promise to advance our understanding of quantum many-body systems and improve machine learning algorithms using near-term quantum computers. Prominent challenges associated with this family of quantum-classical hybrid algorithms are the control of quantum entanglement and quantum gradients linked to their classical optimization. Known as the barren plateau phenomenon, these quantum gradients may rapidly vanish in the presence of volume-law entanglement growth, which poses a serious obstacle to the practical utility of VQAs. Inspired by recent studies of measurement-induced entanglement transition in random circuits, we investigate the entanglement transition in variational quantum circuits endowed with intermediate projective measurements. Considering the Hamiltonian Variational Ansatz (HVA) for the XXZ model and the Hardware Efficient Ansatz (HEA), we observe a measurement-induced entanglement transition from volume-law to area-law with increasing measurement rate. Moreover, we provide evidence that the transition belongs to the same universality class of random unitary circuits. Importantly, the transition coincides with a "landscape transition" from severe to mild/no barren plateaus in the classical optimization. Our work may provide an avenue for improving the trainability of quantum circuits by incorporating intermediate measurement protocols in currently available quantum hardware.


# 1    Introduction

Controlling quantum entanglement has been identified as a critical element in the development of quantum computing. A prominent example where this is of importance is in variational quantum algorithms (VQAs) [1]. VQAs are hybrid quantum-classical algorithms where a parametrized quantum circuit is used to evaluate a cost function. A classical optimizer is then used to find the optimal parameters of the circuit. The performance of such algorithms is impacted by how entanglement is used in the circuit, which often relies on the choice of circuit ansatz and parameter initialization: one must make sure that the proposed circuit is expressive enough, while retaining trainability.

Recently, significant progress has been made in understanding the the evolution of quantum entanglement in random unitary quantum circuits undergoing intermediate projective measurements. In these circuits, random nearest neighbor two-qubit gates locally entangle qubits, which generally leads to volume-law entanglement growth. When such a system is measured at randomly selected locations throughout the circuit, the measured subsystems become disentangled from the rest of the state. One might expect that this leads to a simple decrease in the coefficient of the entanglement growth volume law. However, the competition between local entanglement creation and non-local disentanglement induces a phase transition in the entanglement growth from a volume to an area law at a critical measurement rate $p_c$ [2–12]. Moreover, it appears that this critical behavior is universal, independent of the specific implementation of both the unitary or measurement dynamics. A significant amount of theoretical understanding has been gained about the properties of entanglement phase transitions in random unitary circuits [7,13] by mapping such systems to well-defined statistical mechanics models.

In this work, we connect two highly active fields of research in condensed matter theory and variational quantum computing, by showing that measurement-induced entanglement phase transitions take place in two prototypical variational quantum circuits used within the Variational Quantum Eigensolver (VQE) algorithm [14]. This VQA is used throughout the literature to approximate quantum many-body ground states [15–19], perform quantum chemistry simulations [20–24] or in quantum machine learning approaches [25–28]. Our motivation to

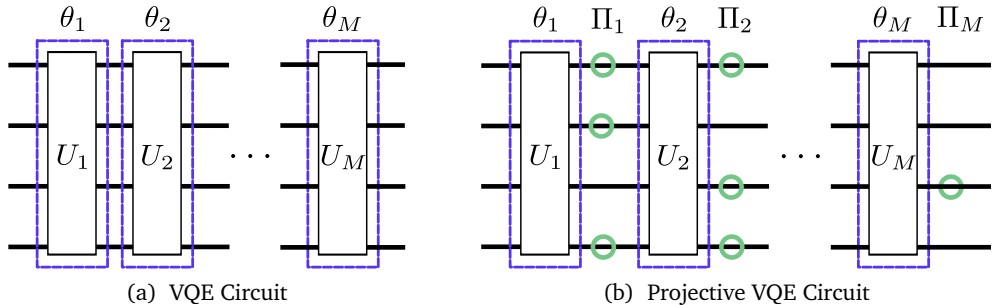

Figure 1: (a) Schematic representation of the circuit $U(\boldsymbol{\theta})$. Each layer $U_m(\theta_m)$ can consist of multiple gates with multiple parameters, hence $\theta_m$ is a vector of parameters. (b) For a circuit undergoing projective measurements, we apply a projector $\Pi_m$ to all qubits. Whether we apply a measurement (green circle) or not is determined by flipping a coin with probability $p$. Once we decide that a measurement will be applied, we sample the projector according to the quantum probability $\tilde{p}_{m,i} = \text{Tr}\{\Pi_{m,i}\rho\}$.

investigate the measurement-induced entanglement transitions in variational quantum circuits are twofold. First, most of the quantum ground states of interacting many-body systems follow the area-law entanglement (up to a logarithmic correction). However, ballistic growth of entanglement in time evolution implies that circuits used in VQE can rapidly develop much more entanglement than what may be needed to simulate these ground states of interest [18,29,30]. Secondly, it is known that randomly initialized variational quantum circuits tend to approximate unitary 2-designs, which are known to have exponentially decaying gradients with respect to the gate parameters as a function of system size. These so-called barren plateaus pose a significant hurdle for variational quantum algorithms, since the number of measurements required to accurately estimate the gradients quickly becomes intractable [31–33]. It has been shown that there is a close relation between entanglement production in a circuit and barren plateaus, hence it is natural to consider constraining the amount of entanglement during parts of the variational optimization as a useful strategy for increasing the trainability of variational circuits [34–36]. We anticipate that the inclusion of interspersed measurements in variational quantum circuits may offer a way to control their quantum entanglement, which could be used as a strategy to overcome barren plateaus. Quantum hardware that allows for intermediate measurements can potentially be used to test these ideas in practice [37–39].

Below we numerically show that the measurement-induced entanglement phase transition takes place in the variational quantum circuits, and coincides with a "landscape transition", a change from a landscape with severe barren plateaus to a landscape with mild or no barren plateaus. This suggests that VQE with intermediate projective measurements can potentially be used to avoid barren plateaus and improve current optimization strategies. In deriving our results, we also provide a modified parameter shift rule for calculating the quantum gradients with intermediate projective measurements that may lead to the development of such algorithms.

## 2 The Variational Quantum Eigensolver

VQE is a hybrid quantum-classical algorithm where we consider a quantum circuit $U(\boldsymbol{\theta})$ parametrized by a set of parameters $\boldsymbol{\theta}$ [14]. We consider an $N$-qubit circuit consisting of

$M$ layers,

$$U(\boldsymbol{\theta}) = \overset{\overset{M}{\longleftarrow}}{\prod_{m=1}} U_m(\theta_m), \tag{1}$$

where $\overset{\longleftarrow}{\prod}$ indicates that the product is ordered from right to left and $\boldsymbol{\theta} = (\theta_1, \ldots, \theta_M)$ are the parameters in each layer. The layers $U(\theta_m)$ can consist of multiple gates, hence $\theta_m$ is a vector consisting of all parameters in layer $m$, see also Fig. 1a. Consider an initial state $\rho_0 = |0\rangle\langle0|$ to which we apply the circuit of Eq. (1). We can calculate the expectation value of a Hermitian operator $H$, or Hamiltonian as

$$\langle H \rangle_{\boldsymbol{\theta}} = \text{Tr}\left\{ U(\boldsymbol{\theta})\rho_0 U^\dagger(\boldsymbol{\theta})H \right\}. \tag{2}$$

By invoking the variational principle,

$$E_{\text{ground}} \le E(\boldsymbol{\theta}) = \langle H \rangle_{\boldsymbol{\theta}}, \tag{3}$$

one can use a classical optimization routine to minimize the variational energy $E(\boldsymbol{\theta})$ given a Hamiltonian $H$ with respect to the parametrized wave function $\rho(\boldsymbol{\theta}) = U(\boldsymbol{\theta})\rho_0 U^\dagger(\boldsymbol{\theta})$ and approximate the ground state of $H$.

As with other variational methods, the choice of ansatz $U(\boldsymbol{\theta})$ is crucial since the ground states must be reachable from the initial state by application of this unitary. There exists a variety of proposals, including the so-called Hamiltonian Variational Ansatz (HVA) [15–19] and the Hardware Efficient Ansatz (HAE) [20, 40, 41]. The former exploits the structure of the Hamiltonian for the unitary ansatz design, whereas the latter aims to provide a hardware-friendly parametrization with enough degrees of freedom to capture a variety of states.

## 3  Measurement–induced entanglement phase transitions

We are interested in studying random ensembles of typical VQE circuits undergoing projective measurements and the entanglement properties of the states they produce. Given a state $\rho$, a projective measurement in the computational basis results in

$$\rho' = \frac{\Pi_i \rho \Pi_i}{\text{Tr}\{\Pi_i \rho\}}, \tag{4}$$

where $\Pi_i = |i\rangle\langle i|$ are the projectors onto the $\sigma^z$ basis. Which projector $\Pi_i$ is applied depends on the quantum probability $\text{Tr}\{\Pi_i \rho\}$.

Consider the circuit in Eq. (1). After each layer $m$, with probability $p$ (the measurement rate), we apply a projective measurement to each qubit. For $M$ layers, we then obtain the variational state

$$\rho_M(\boldsymbol{\theta}) = \left( \overset{\overset{M}{\longleftarrow}}{\prod_{m=1}} \Pi_m U_m(\theta_m) \right) \rho_0 \left( \overset{\overset{M}{\longrightarrow}}{\prod_{m=1}} U_m^\dagger(\theta_m)\Pi_m \right) p_M^{-1}(\boldsymbol{\theta}), \tag{5}$$

where $p_M(\boldsymbol{\theta})$ is the probability of obtaining the state $\rho_M(\boldsymbol{\theta})$ given the $M$ sets of measurements performed, see also Fig. 1b. The projective measurement is represented by the projector $\Pi_m = \Pi_{m,0} \otimes \ldots \otimes \Pi_{m,N}$ where $\Pi_{m,i} \in \{|0\rangle\langle0|, |1\rangle\langle1|\}$ if we perform a measurement and $\Pi_{m,i} = I$ otherwise. Here, $\rho_M(\boldsymbol{\theta})$ is the normalized state obtained after applying the circuit with intermediate measurements. Each projector $\Pi_m$ has $3^N$ different configurations, hence there will be a total of $3^{NM}$ possible states $\rho_M(\boldsymbol{\theta})$. Note that each state $\rho_M(\boldsymbol{\theta})$ corresponds

to a pure state. Also, we want to emphasize that we are not performing any optimization; we consider the variational circuit at initialization.

Given a state produced by quantum circuit interspersed with intermediate measurements, we can calculate the bipartite von Neumann entanglement entropy $S(N, p)$ between two halves of the system as a function of the measurement rate $p$. These measurements disentangle the system over any length scale due to a local projection onto a single state. As a result, the unitary dynamics locally entangles nearest neighbor qubits, whereas measurements globally remove entanglement between different subsystems. This competition induces a dynamical phase transition between a volume and area law regime of entanglement scaling at a critical measurement rate $p_c$ [2–12]. Although the critical point $p_c$ can vary between different types of random unitary dynamics and measurement schemes, the critical exponent characterizing the correlation length scale divergence $\xi \propto (p - p_c)^{-\nu}$ appears to be the same for different models at $\nu \approx 4/3$. This critical exponent can be derived by considering toy models and mapping the projective dynamics to a two-dimensional percolation model, which is exactly solvable [3, 4, 7, 8, 13].

Central to the investigations on phase transitions induced by measurements is the concept of steady state entanglement dynamics [3–5]. Given a circuit with a number of qubits $N$, we are primarily interested in the late time behavior when $M \to \infty$. In this infinite depth (long time) limit we expect the system to evolve into a steady state, characterized by a typical value of entanglement entropy that depends on the measurement rate $p$, but not the dynamics at finite times. In order to characterize this regime, we can investigate the average entanglement entropy as a function of depth for different values of $p$. For the moderate system sizes considered in this work, we observe steady state entanglement dynamics at $M = 16$.

For our numerical study, we investigate the projective entanglement dynamics of the XXZ-chain HVA and the HEA, whose circuits are depicted in Fig. 2. Notice that the dynamics in the HVA is specified by a Hamiltonian in contrast to random unitaries. The HVA for the XXZ model is of particular interest since the XXZ Hamiltonian is Bethe-ansatz integrable, i.e. there exists an analytical solution for the energy spectrum. Additionally, the entanglement properties of these systems undergoing quenches can be understood analytically [42, 43]. For such models, it is still an open question if the corresponding unitary dynamics interspersed with measurements will produce a measurement-induced entanglement phase transitions [7]. Here, we address a closely related model, where the unitary dynamics are generated by random quenches under the XXZ Hamiltonian. For the HEA, we expect that the behavior is close to that of random circuits [32].

Since phase transitions only occur in the thermodynamic limit $N \to \infty$, we have to take care of the finite-size effects in analyzing our numerical data. To account for finite-size effects, we fit the scaling form [3, 4, 7]

$$S(N, p, \nu) - S(N, p_c, \nu) = f(N^{1/\nu}(p - p_c)), \tag{6}$$

where $f$ is a scaling function, to get a data collapse of the individual circuits of size $N$. To determine $p_c$ and $\nu$, we minimize a Chi-squared statistic between the scaling form above and the data, and use a statistical bootstrap to verify the integrity of the fit. In Fig. 3 we find critical exponents close to the previously mentioned value of $\nu \approx 4/3$. To extrapolate the critical exponent to the thermodynamic limit, we do a linear fit of $\nu$ as a function of $1/N'$ where $N_{max}/2 \leq N' \leq N_{max}$ is the largest value of $N$ in the data set. The intercept then gives us the value of $\nu$ for $N' \to \infty$ [3]. The details of our statistical estimation procedure are outlined in App. A. In addition to the finite scaling analysis, we can investigate the quantum mutual information,

$$I(A, B) = S_A(N, p) + S_B(N, p) - S_{A \cup B}(N, p), \tag{7}$$

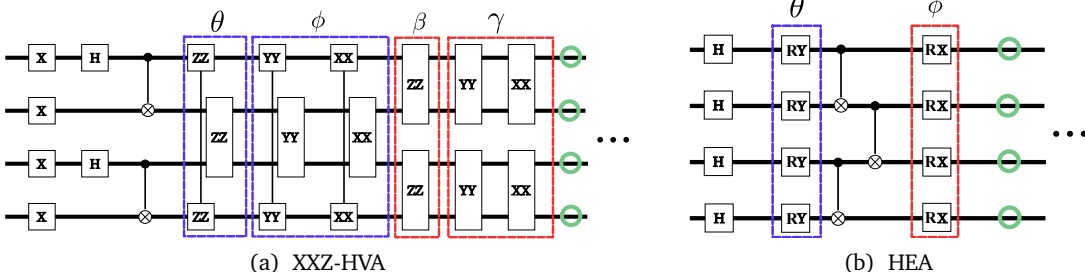

Figure 2: Schematic depiction of the circuits studied. (a) For the XXZ-HVA, we prepare a Bell state on the even sites and alternatingly apply ZZ, YY and XX two-qubit rotations on odd and even bonds in the chain, which corresponds to the unitary rotations generated by the terms in the of the Hamiltonian $H_{\text{XXZ}} = \sum_{i=1}^{N} \left[ \sigma_i^x \sigma_{i+1}^x + \sigma_i^y \sigma_{i+1}^y + \Delta \sigma_i^z \sigma_{i+1}^z \right]$. For the odd (even) bonds, the ZZ rotations are parametrized by $\theta$ ($\beta$) whereas the YY and XX rotations are parametrized by $\phi$ ($\gamma$). Hence the gates in the circuit are $U_{zz}^{\text{odd}}(\theta) = \exp\{-i\theta\sigma_i^z\sigma_{i+1}^z\}$, $U_{zz}^{\text{odd}}(\phi) = \exp\{-i\phi\sigma_i^z\sigma_{i+1}^z\}$, $U_{xx+yy}^{\text{even}}(\beta) = \exp\{-i\beta(\sigma_i^x\sigma_{i+1}^x + \sigma_i^y\sigma_{i+1}^y)\}$ and $U_{xx+yy}^{\text{even}}(\gamma) = \exp\{-i\gamma(\sigma_i^x\sigma_{i+1}^x + \sigma_i^y\sigma_{i+1}^y)\}$. (b) The initial state in the HEA consists of the equal superposition followed by $L$ layers of low-depth entangling unitaries. These unitaries consists of $N$ Pauli-Y rotations on each qubit, a chain of nearest neighbor CNOTs and $N$ Pauli-X rotations on each qubit. All $2N$ rotations are controlled by individual parameters $\theta_{i,l}, \phi_{i,l}$, where $i = 1, \ldots, N$ indicates the qubit and $l = 1, \ldots, M$ indicates the layer. After each layer, we perform a projective measurement according to Eq. (4) with probability $p$ on each qubit (indicated by the green circles here), bringing the average number of measurements in the circuits to $NMp$.

between qubits $A$ and $B$ separated by a distance $r$, which we expect to peak at a critical point due to subsystem correlations becoming non-negligible. From these data, we find similar critical measurement rates $p_c \approx 0.25$ and $p_c \approx 0.5$ for the XXZ-HVA and HEA, respectively. In App. B. we give further details on this procedure.

The results in Fig. 3 suggest that an entanglement phase transition takes place in two prototypical circuits used in VQAs. Although we have studied static circuits here where no optimization takes place, we can investigate how the projective measurements affect the gradients with respect to the gate parameters in the circuit.

## 4 Projective gradients and barren plateaus

The variational energy in Eq. (3) is typically a non-convex function of the gate parameters $\boldsymbol{\theta}$. In practice one typically uses a gradient-based method to find a minimum of the cost function. To calculate the necessary gradients with respect to the layer parameters, we can employ hardware-friendly methods, most of which rely on the usage of the so-called parameter-shift rule [44–51]. In its standard form, this allows one to calculate the gradient with respect to

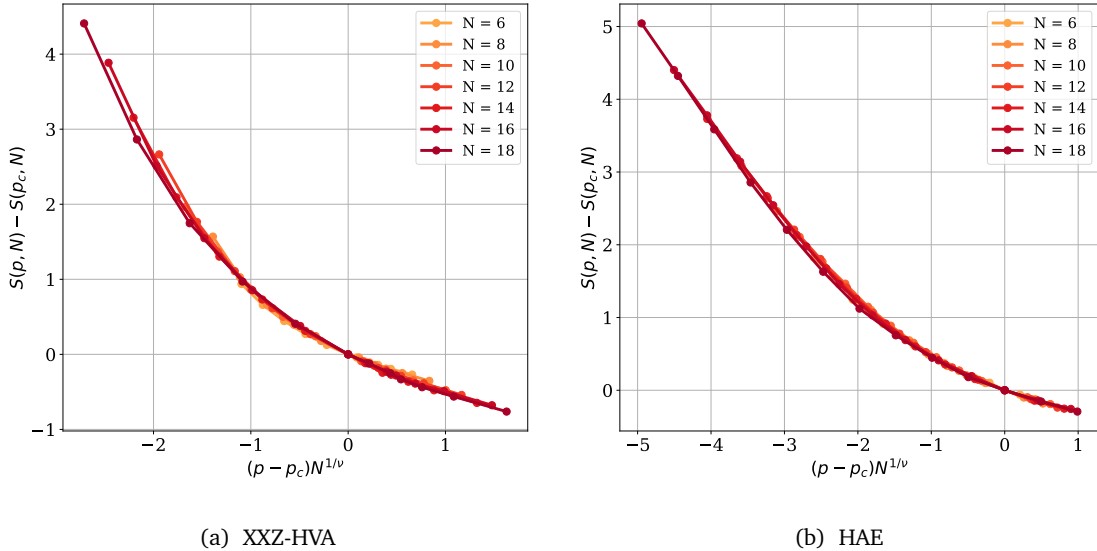

(a) XXZ-HVA

(b) HAE

Figure 3: Data collapse of the average entanglement entropies. (a) For the XXZ-HVA, we find $p_c = 0.25$ and $\nu \approx 1.22 \pm 0.24$. (b) For the HEA, we find $p_c \approx 0.5$ and $\nu \approx 1.26 \pm 0.23$. The error bars are calculated as the difference between the critical exponent in the thermodynamic extrapolation and the finite-size data collapse. The average $S(p,N)$ is obtained by averaging over $3 \times 10^3$ circuit realizations with all circuit parameters sampled uniformly in $(0, 2\pi)$. After each layer, we apply a computational basis measurement with probability $p$. Due to the difficulty in simulating large systems, we restrict ourselves to $N = 6, 8, \ldots, 18$.

the parameters of a gate generated by a Pauli operator as

$$\partial_{\theta_l} \langle H \rangle_{\boldsymbol{\theta}} = \frac{1}{2} \operatorname{Tr} \left\{ \left( U(\theta_1, \ldots, \theta_l + \frac{\pi}{2}, \ldots, \theta_M) \rho_0 U^\dagger(\theta_1, \ldots, \theta_l + \frac{\pi}{2}, \ldots, \theta_M) \right. \right. \tag{8}$$

$$+ U(\theta_1, \ldots, \theta_l - \frac{\pi}{2}, \ldots, \theta_M) \rho_0 U^\dagger(\theta_1, \ldots, \theta_l - \frac{\pi}{2}, \ldots, \theta_M) \Big) H \Big\} \tag{9}$$

$$= \frac{1}{2} \left( \langle H \rangle_{\boldsymbol{\theta}}^{+,l} - \langle H \rangle_{\boldsymbol{\theta}}^{-,l} \right). \tag{10}$$

Where we use $\langle . \rangle_{\boldsymbol{\theta}}^{\pm,l}$ to denote the expectation value under a circuit $U(\boldsymbol{\theta})$ where parameter $l$ has been shifted by $\pm \pi/2$. In other words, the gradient can be calculated by shifting the parameter $\theta_l$ by $\pm \pi/2$ and calculating the difference of the expectation values of $H$ under the shifted circuits. Unfortunately, this kind of gradient calculation is plagued by barren plateaus in the cost landscape: gradients with respect to the gate parameters vanish exponentially with the number of qubits in the circuit, preventing us from optimizing the circuit. To mitigate this problem, a variety of recent works are aimed at finding ways to avoid these regions where optimization is hard [32, 35, 52–56].

Here, we investigate the barren plateau problem under the influence of projective measurements, more specifically the variance of the gradients in the XXZ-HVA and the HEA with intermediate projective measurements. There has been prior work on gradient through non-unitary quantum circuits. For instance, in [57] the quantum natural gradient [58] is extended to quantum channels. Additionally, in [59] measurement-based VQE is investigated, but only in the context where an entangled cluster state is prepared and measurements are directly part of the algorithm [60]. None of these works consider quantum gradients through a circuit

undergoing projective measurements, which is the case we consider here.

The gradient with respect to a single parameter $\theta_l$ of the expectation value of a Hermitian operator $H$ undergoing a set of measurements is given by

$$\partial_{\theta_l} \langle H_M \rangle_{\boldsymbol{\theta}} = \partial_{\theta_l} \text{Tr}\{H\rho_M(\boldsymbol{\theta})\}, \tag{11}$$

where $\rho_M(\boldsymbol{\theta})$ is given in Eq. (5)). In App. C, show that the full projective gradient is can be written as

$$\partial_{\theta_l} \langle H_M \rangle_{\boldsymbol{\theta}} = \frac{1}{2}\left( \left(\langle H \rangle_{\boldsymbol{\theta}}^+ - \langle H \rangle_{\boldsymbol{\theta}}\right) \frac{p_M^{+,l}}{p_M} - \left(\langle H \rangle_{\boldsymbol{\theta}}^- - \langle H \rangle_{\boldsymbol{\theta}}\right) \frac{p_M^{-,l}}{p_M} \right). \tag{12}$$

The probabilities $p_M$ and $p_M^{\pm,l}$ are the probabilities of obtaining $\rho_M(\boldsymbol{\theta})$ and $\rho_M(\theta_1, \ldots, \theta_l \pm \pi/2, \ldots \theta_M)$, respectively. Similarly, the expectation values $\langle H \rangle_{\boldsymbol{\theta}}$ and $\langle H \rangle_{\boldsymbol{\theta}}^{\pm}$ correspond to the expectation value of $H$ under $\rho_M(\boldsymbol{\theta})$ and $\rho_M(\theta_1, \ldots, \theta_l \pm \pi/2, \ldots \theta_M)$, respectively. Note that obtaining these probabilities will be difficult and require a large number of measurements, since estimating the ratio $p_M^{-,l}/p_M$ requires full knowledge of the wave function.

To investigate the severity of the barren plateau effect, we consider the same circuits as in Fig. 3 and examine the projective gradients of Eq. (12) with respect to the expectation value of $H = Z_0 Z_1$. We calculate the projective gradients for the first circuit parameter ($\theta_1$ in the first parametrized layer in both the HVA and HEA (see Fig. 2). We consider a depth $M = 16$ circuit for system sizes $N = 8, \ldots 18$. In Fig. 4a and Fig. 4b, we observe that the gradient variances in both the XXZ-HVA and HEA transition from exponentially decaying to a constant as the measurement rate increases. This transition coincides with the critical measurement rate for the volume-area law transition (See App. E). Therefore, we see that the measurement-induced entanglement phase transitions induces a landscape transition in the circuit from mild/severe barren plateaus to no barren plateaus.

This landscape transition can serve as the motivation for a projective gradient VQE algorithm where the early optimization of the circuit is done with projective gradients to escape barren plateaus due to initialization. However, calculating the gradients in Eq. (12) is exponentially hard in the number of layers $M$, since we need accurate estimates of $p_M$ and $p_M^{\pm,l}$.

On the contrary, the mixture of all pure states $\rho_M(\boldsymbol{\theta})$ has a simple gradient formula that can be calculated in practice, as we show in App. D. The resulting ensemble however, corresponds to system at infinite temperature [3–5]. It is known that such a high temperature ensemble will suffer again from barren plateaus [33]. Additionally, we require a pure state as the outcome of our optimization algorithm, which will require annealing the measurement rate to zero during the optimization. Any useful variational algorithm with intermediate measurements must not remix all projective states but still be efficiently calculable. We leave the exploration of this class of algorithms as future work.

## 5  Outlook

In this work, we demonstrated the existence of a measurement-induced entanglement phase transition in variational quantum circuits which coincides with a "landscape transition" in the behavior of quantum gradients. As mentioned earlier, the exponentially-vanishing quantum gradients in presence of volume-law entanglement growth, the so-called barren plateau, is a serious obstacle in the applications of variational quantum circuits. Our work suggests that intermediate projective measurements may provide a useful knob to control the barren plateau issue. Inclusion of the measurement protocol in the quantum-classical hybrid algorithm would be a timely development given that quantum computing hardware companies like IBM and

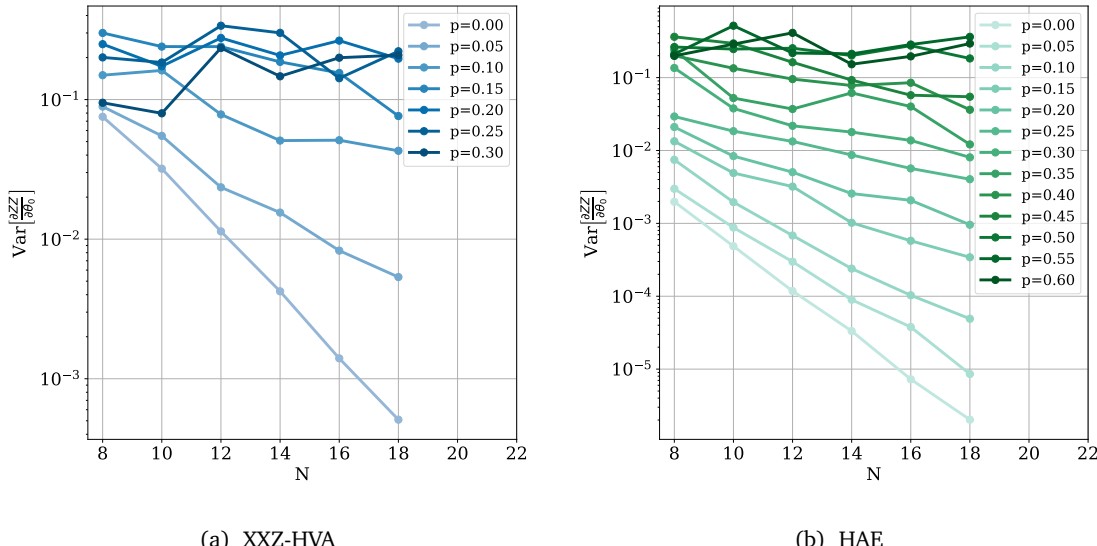

(a) XXZ-HVA

(b) HAE

Figure 4: Variance of the projective gradients taken with respect to the first parameter in the circuit ($\theta$ in the first parametrized layer in both the HVA and HEA, see Fig. 2). The variances are estimated over $10^3$ samples where for each data point, we randomly choose measurements with probability $p$ and uniformly sample the gate parameters. The gradient is then calculated exactly from Eq. (12). We emphasize that these gradients are thus calculated with respect to the individual pure states resulting from measuring the state during the application of the circuit. For the 1D HVA-XXZ circuits with depth $M = 16$ (a) and the 1D HAE circuit with depth $M = 16$ (b) the gradient variance becomes constant as the measurement rate $p$ increases.

Honeywell now allow their users to perform mid-circuit measurements, enabling the real-time logic required for performing these algorithms in an experimental setting [37–39]. In particular, the Hamiltonian variational quantum circuits considered in this work could be implemented in the quantum hardware. For the projective gradient VQE, the exponential sum in Eq. (12) currently inhibits the number of measurements that can be performed in practice. A detailed analysis of when and how a projective circuit optimization can be practical and "advantageous" would be an excellent topic of future study.

For a practical implementation of a projective gradient VQE algorithm, note that the scheme we provided here is quite general and many extensions and modifications are possible. For instance, the projective measurements used in this work can be replaced by general Positive Operator Value Measures (POVM) or parametrized measurements. Additionally, we have focused on one-dimensional quantum circuits where the measurement-induced entanglement transition belongs to the same universality class as in the random unitary circuits. It would be interesting to consider moderately sized quantum circuits with a two-dimensional topology, and see if a similar phase transition appears there and investigate the universality class.

## Acknowledgements

We would like to thank Henry Yuen for the discussions during the course of this project. Y.B.K. is supported by the NSERC of Canada and the Center for Quantum Materials at the University of Toronto. J.C. acknowledges support from NSERC, the Shared Hierarchical Academic Research

Computing Network (SHARCNET), Compute Canada, Google Quantum Research Award, and the CIFAR AI chair program. Resources used in preparing this research were provided, in part, by the Province of Ontario, the Government of Canada through CIFAR, and companies sponsoring the Vector Institute www.vectorinstitute.ai/#partners. C.Z. acknowledges support from the U.S. National Science Foundation under Grant No. 2116246 and the U.S. Department of Energy, Office of Science, National Quantum Information Science Research Centers, Quantum Systems Accelerator, and the Postgraduate Affiliate Award from the Vector Institute.

## A  Finite-scaling analysis and data collapse

The correlation length $\xi$ of a system quantifies the length scale over which parts of a system are correlated. When a system undergoes a continuous phase transition, the correlation length diverges. Phase transitions only occur in the thermodynamic limit, and hence simulations of finite-sized systems will contain artifacts that have to be accounted for in order to capture the correct behavior [61]. In particular, for a finite system the correlation length $\xi$ cannot become infinite and is cut off at $L^d$, the maximum volume of a finite $d$-dimensional system. To account for this effect, we can perform a finite-scaling analysis.

The entanglement entropy as a function of measurement rate is conjectured to follow a volume law for $p < p_c$, a constant plus logarithmic correction at $p = p_c$ and area law for $p > p_c$ [3,4,7]. We can therefore construct a scaling form of the entanglement entropy as

$$S(N, p, \nu) = S(N, p_c, \nu) + f\left(N^{1/\nu}(p - p_c)\right), \tag{A.1}$$

where $S(N, p, \nu)$ denotes the von Neumann entropy at measurement rate $p$ and $f$ is a scaling function. The critical exponent $\nu$ determines the scaling of the entanglement entropy near $p_c$. If this scaling form is correct, we should be able to account for finite-size effects and all the data can be appropriately rescaled to match a single curve representing $f$ with a proper choice of $\nu$.

To determine the critical exponents, we fit a 5th-degree polynomial $g$ to our data using a Nelder-Mead optimization [62] and minimize the $\chi^2$-statistic

$$\chi^2 = \sum_i \frac{(S(N_i, p_i, \nu) - \tilde{S}(N_i, p_i, \nu))^2}{\Delta S}. \tag{A.2}$$

Here, $\tilde{S}(N_i, p_i, \nu)$ is estimated from the data and $S(N_i, p_i, \nu)$ is the proposed scaling form from Eq. (A.1). $\Delta S$ is the standard deviation of the von Neumann entropies which arises due to the fluctuations induced by the randomized measurements and their outcomes. From the unscaled data, we determine a set of potential critical points $p_c$ and fit the above $\chi^2$-statistic to determine $\nu$. We then report the values of $p_c$ and $\nu$ that provided the best fit.

To verify the stability of the fit, we perform a statistical bootstrapping procedure to estimate the error bars on the fitted critical exponent $\nu$. We take $K_{\text{boot}} = 100$, where each data set consists of $K$ samples obtained by sampling from the entire data set of $3 \times 10^3$ data points with replacement. The final obtained error bars on $\nu$ are $\approx 0.01$.

We can extrapolate our result to the thermodynamic limit by fitting the data for $N' = N_{\text{max}}/2$ to $N' = N_{\text{max}}$ and plotting the resulting values for $\nu$ against $1/N'$ [3]. By doing a linear fit on the resulting data, we obtain

$$\tilde{\nu}(N') = a\frac{1}{N'} + b, \tag{A.3}$$

and so the intercept $b$ corresponds to the value of $\nu$ in the thermodynamic limit, since $\lim_{N' \to \infty} 1/N' = 0$. When fitting the data, we weigh the errors by the standard errors obtained in the statistical bootstrap described above.

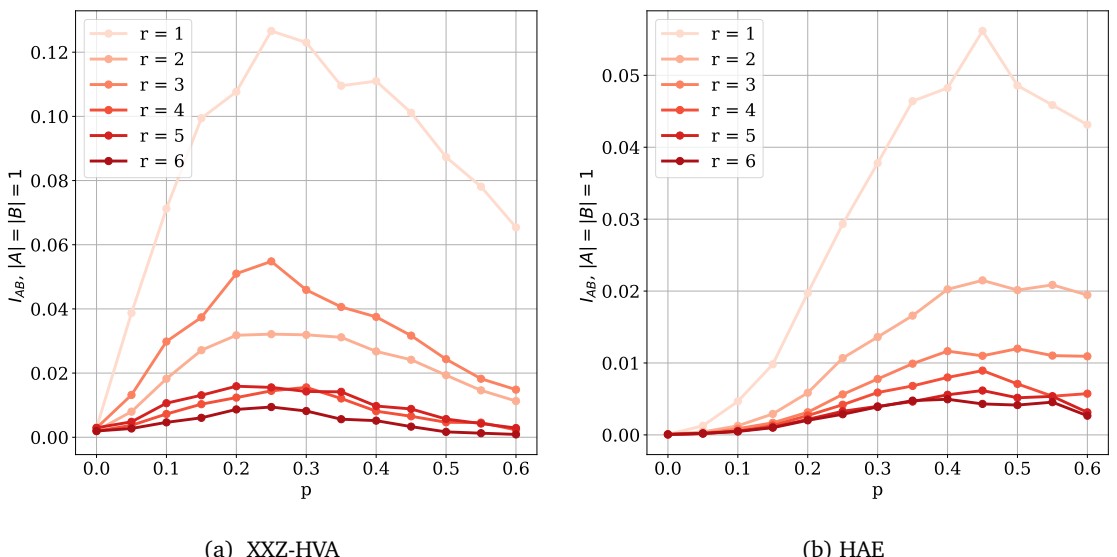

(a) XXZ-HVA

(b) HAE

Figure B1: Quantum mutual information between two qubits $A$ and $B$ separated by a distance $r$ on a chain of length 16. The mutual information is averaged over $3 \times 10^3$ samples, where each sample corresponds to a random circuit realization, as described in the main text.

# B  Mutual information

The quantum mutual information can be used to quantify subsystem correlations, and subsequently detect phase transitions since we expect correlations to divergence at criticality [3,4,6]. As additional confirmation that the critical values $p_c$ estimated from the prior analysis are correct, we calculate the quantum mutual information as,

$$I(A,B) = S_A(N,p) + S_B(N,p) - S_{A \cup B}(N,p). \tag{B.1}$$

Here, we take the same approach as in [4], and take $A$ and $B$ to be two single qubit subsystems $|A| = 1$ and $|B| = 1$. We then vary the distance $r$ between qubit $A$ and $B$, to determine the effect of the distance on the subsystem correlations. In Fig. B1, we observe two broad peaks around the previously found values $p_c \approx 0.25$ and $p_c \approx 0.5$ for the XXZ-HVA and HAA, respectively.

# C  Projective gradients

Let $|\psi\rangle$ be a quantum state of an $n$-qubit system with corresponding density operator $|\psi\rangle\langle\psi| = \rho \in L(\mathbb{C}^{2^n})$. We consider a circuit of $M$ layers $U_m$, where each layer has a set of parameters $\theta_m \in \mathbb{R}^{d_m}$ and $d_m$ is the number of parameters in that layer. We can then write a parametrized state as

$$|\psi(\theta_{1:k})\rangle = \overleftarrow{\prod_{m=1}^{M}} U_m(\theta_k) |0\rangle , \tag{C.1}$$

where $\overleftarrow{\prod}$ indicates that the product is ordered from right to left. A projective measurement transforms a state as

$$\rho \mapsto \frac{\Pi \rho \Pi}{\text{Tr}\{\Pi \rho\}} , \tag{C.2}$$

where $\Pi$ is a projector onto an eigenbasis a Hermitian observable and is therefore Hermitian itself. Since $\Pi$ is a projector it satisfies $\Pi^2 = \Pi$. The normalization constant $p = \mathrm{Tr}\{\Pi\rho\}$ gives the overlap of the state $\rho$ with the basis onto which $\Pi$ projects the state.

We consider the case where after each layer $U_m(\theta_k)$ we apply a projective measurement $\Pi_m$, where $\Pi_m = \Pi_{m,0} \otimes \ldots \otimes \Pi_{m,N}$ with $\Pi_{m,i} \in \{|0\rangle\langle0|, |1\rangle\langle1|, I\}$. We denote by $\rho_M$ the state resulting from applying $M$ projectors $\Pi$ to the circuit. Similarly, we denote by $p_M$ the probability of obtaining the state $\rho_M$.

Consider an initial state $\rho_0 = |0\rangle\langle0|^{\otimes N}$, to which we apply the unitary $U_1(\theta_1)$ followed by a projective measurement $\Pi_1$,

$$\rho_1(\theta_1) = \frac{\Pi_1 U_1(\theta_1)\rho_0 U_1^\dagger(\theta_1)\Pi_1}{\mathrm{Tr}\{\Pi_1 U_1(\theta_1)\rho_0 U_1^\dagger(\theta_1)\Pi_1\}} \tag{C.3}$$

$$= \frac{\Pi_1 U_1(\theta_1)\rho_0 U_1^\dagger(\theta_1)\Pi_1}{p_1(\theta_1)}. \tag{C.4}$$

Next, we add an additional unitary and measurement,

$$\rho_2(\theta_1, \theta_2) = \frac{\Pi_2 U_2(\theta_2)\rho_1(\theta_1) U_2^\dagger(\theta_2)\Pi_2}{\mathrm{Tr}\{\Pi_2 U_2(\theta_2)\rho_1(\theta_1) U_2^\dagger(\theta_2)\Pi_2\}} \tag{C.5}$$

$$= \frac{\Pi_2 U_2(\theta_2)\Pi_1 U_1(\theta_1)\rho_0 U_1^\dagger(\theta_1)\Pi_1 U_2^\dagger(\theta_2)\Pi_2}{\mathrm{Tr}\{\Pi_2 U_2(\theta_2)\Pi_1 U_1(\theta_1)\rho_0 U_1^\dagger(\theta_1)\Pi_1 U_2^\dagger(\theta_2)\Pi_2\}} \times \frac{p_1(\theta_1)}{p_1(\theta_1)} \tag{C.6}$$

$$= \frac{\Pi_2 U_2(\theta_2)\Pi_1 U_1(\theta_1)\rho_0 U_1^\dagger(\theta_1)\Pi_1 U_2^\dagger(\theta_2)\Pi_2}{p_2(\theta_1, \theta_2)}. \tag{C.7}$$

Note how the normalization constant of $\rho_1(\theta_1)$ cancels. Generalizing this to $M$ projectors, we get the general form

$$\rho_M(\theta_1, \ldots, \theta_M) = \left(\overset{\overleftarrow{M}}{\prod_{m=1}} \Pi_m U_m(\theta_m)\right) \rho_0 \left(\overset{\overrightarrow{M}}{\prod_{m=1}} U_m^\dagger(\theta_m)\Pi_m\right) p_M^{-1}(\theta_1, \ldots, \theta_M) \tag{C.8}$$

$$= \tilde{\rho}_M(\theta_1, \ldots, \theta_M) p_M^{-1}(\theta_1, \ldots, \theta_M), \tag{C.9}$$

where

$$\tilde{\rho}_M(\theta_1, \ldots, \theta_M) = \left(\overset{\overleftarrow{M}}{\prod_{m=1}} \Pi_m U_m(\theta_m)\right) \rho_0 \left(\overset{\overrightarrow{M}}{\prod_{m=1}} U_m^\dagger(\theta_m)\Pi_m\right), \tag{C.10}$$

$$p_M(\theta_1, \ldots, \theta_M) = \mathrm{Tr}\{\tilde{\rho}_M(\theta_1, \ldots, \theta_M)\} \tag{C.11}$$

are the unnormalized state and its normalization constant, respectively. To simplify the notation, we will write $\boldsymbol{\theta} \equiv (\theta_1, \ldots, \theta_M)$.

We are interested in the derivative of an expectation value

$$\langle H_M \rangle_{\boldsymbol{\theta}} \equiv \mathrm{Tr}\{\rho_M(\boldsymbol{\theta})H\}, \tag{C.12}$$

where $H$ is a Hermitian operator. We write the state as the product of an unnormalized state and its normalization constant

$$\mathrm{Tr}\{\rho_M(\boldsymbol{\theta})H\} = \mathrm{Tr}\{\tilde{\rho}_M(\boldsymbol{\theta}) p_M^{-1}(\boldsymbol{\theta})H\}. \tag{C.13}$$

Hence the derivative consists of two parts via the product rule

$$\partial_{\theta_l} \langle H_M \rangle_{\boldsymbol{\theta}} = \overbrace{\mathrm{Tr}\{\partial_{\theta_l} (\tilde{\rho}_M(\boldsymbol{\theta})) p_M^{-1}(\boldsymbol{\theta}) H\}}^{(i)} + \overbrace{\mathrm{Tr}\{\tilde{\rho}_M(\boldsymbol{\theta}) \partial_{\theta_l} (p_M^{-1}(\boldsymbol{\theta})) H\}}^{(ii)} . \tag{C.14}$$

(i) For the derivative of the unnormalized state, we get

$$\mathrm{Tr}\{(\partial_{\theta_l} \tilde{\rho}_M(\boldsymbol{\theta})) H\} \tag{C.15}$$

$$= \langle 0| \left( \overrightarrow{\prod_{m=1}^{M}} U_m^\dagger(\theta_m)\Pi_m \right) H \left( \overleftarrow{\prod_{m=l+1}^{M}} \Pi_m U_m(\theta_m) \right) \Pi_l \partial_{\theta_l} U_l(\theta_l) \left( \overleftarrow{\prod_{m=1}^{l-1}} \Pi_m U_m(\theta_m) \right) |0\rangle$$

$$+ \langle 0| \left( \overrightarrow{\prod_{m=1}^{l-1}} U_m^\dagger(\theta_m)\Pi_m \right) \partial_{\theta_l} U_l^\dagger(\theta_l)\Pi_l \left( \overrightarrow{\prod_{m=l+1}^{M}} U_m^\dagger(\theta_m)\Pi_m \right) H \left( \overleftarrow{\prod_{m=1}^{M}} \Pi_m U_m(\theta_m) \right) |0\rangle$$

$$= \langle \tilde{\psi}_0| U_l^\dagger(\theta_l)\tilde{H}\partial_{\theta_l} U_l(\theta_l) |\tilde{\psi}_0\rangle + \langle \tilde{\psi}_0| \partial_{\theta_l} U_l^\dagger(\theta_l)\tilde{H}U_l(\theta_l) |\tilde{\psi}_0\rangle , \tag{C.16}$$

where

$$|\tilde{\psi}_0\rangle = \left( \overleftarrow{\prod_{m=1}^{l-1}} \Pi_m U_m(\theta_m) \right) |0\rangle \tag{C.17}$$

is an unnormalized state and

$$\tilde{H} = \left( \overrightarrow{\prod_{m=l+1}^{M}} U_m^\dagger(\theta_m)\Pi_m \right) H \left( \overleftarrow{\prod_{m=l+1}^{M}} \Pi_m U_m(\theta_m) \right) . \tag{C.18}$$

If $U(\theta_l)$ is generated by a Pauli operator $A$, then $\partial_{\theta_l} U_l(\theta_l) = -\frac{i}{2} A U_l(\theta_l)$ and so we can use the parameter-shift rule [44, 45]

$$-\frac{i}{2} \langle \tilde{\psi}_0| \left[ A, U_l^\dagger(\theta_l)\tilde{H}U_l(\theta_l) \right] |\tilde{\psi}_0\rangle = \tag{C.19}$$

$$= \frac{1}{2} \left( \langle \tilde{\psi}_0| U^\dagger(\theta_l + \frac{\pi}{2})\tilde{H}U(\theta_l + \frac{\pi}{2}) - U^\dagger(\theta_l - \frac{\pi}{2})\tilde{H}U(\theta_l - \frac{\pi}{2}) |\tilde{\psi}_0\rangle \right), \tag{C.20}$$

where

$$\tilde{\Pi} = \left( \overrightarrow{\prod_{m=l+1}^{M-1}} U_m^\dagger(\theta_m)\Pi_m \right) \Pi_M \left( \overleftarrow{\prod_{m=l+1}^{M-1}} \Pi_m U_m(\theta_m) \right), \tag{C.21}$$

is obtained by setting $H = I$ in Eq. (C.18).

If the expectation values in Eq. (C.20) were with respect to properly normalized states, then this would provide a strategy for measuring the projective gradient. Hence, we need to first normalize the state in order to be able to perform the gradient calculation on the device. The normalization constants for the plus and minus shifted circuits are given by

$$p_M^{\pm,l} \equiv \langle \tilde{\psi}_0| U^\dagger \left( \theta_l \pm \frac{\pi}{2} \right) \tilde{\Pi} U \left( \theta_l \pm \frac{\pi}{2} \right) |\tilde{\psi}_0\rangle . \tag{C.22}$$

Therefore, if we multiply with the identity

$$\mathrm{Tr}\{(\partial_{\theta_l} \tilde{\rho}_M(\boldsymbol{\theta})) H\} = \frac{1}{2} \langle \tilde{\psi}_0| \left( U^\dagger \left( \theta_l + \frac{\pi}{2} \right) \tilde{H} U \left( \theta_l + \frac{\pi}{2} \right) \times \frac{p_M^{+,l}}{p_M^{+,l}} \right. \tag{C.23}$$

$$\left. - U^\dagger \left( \theta_l - \frac{\pi}{2} \right) \tilde{H} U \left( \theta_l - \frac{\pi}{2} \right) \times \frac{p_M^{-,l}}{p_M^{-,l}} \right) |\tilde{\psi}_0\rangle \tag{C.24}$$

$$= \frac{1}{2} \left( \langle H \rangle_{\boldsymbol{\theta}}^{+,l} p_M^{+,l} - \langle H \rangle_{\boldsymbol{\theta}}^{-,l} p_M^{-,l} \right) . \tag{C.25}$$

Here, $\langle H \rangle_{\boldsymbol{\theta}}^{\pm,l}$ is the expectation value of the observable $H$ after the measurements $\{\Pi_1, \ldots, \Pi_M\}$ have been applied and parameter $\theta_l$ has been shifted by $\pm \pi/2$.

(ii) For the gradient of the inverse of the normalization constant, we get

$$\text{Tr}\{\tilde{\rho}_M(\boldsymbol{\theta})(\partial_{\theta_l} p_M^{-1}(\boldsymbol{\theta}))H\} = -\langle H_M \rangle_{\boldsymbol{\theta}}\, p_M^{-1}(\boldsymbol{\theta}) \partial_{\theta_l} p_M(\boldsymbol{\theta}), \tag{C.26}$$

where we used the normalization constant to write $\text{Tr}\{\tilde{\rho}_M(\boldsymbol{\theta}) p_M^{-1}(\boldsymbol{\theta})H\} = \langle H_M \rangle_{\boldsymbol{\theta}}$, the expectation value of $H$ with respect to the measured circuit. The final step is to calculate $\partial_{\theta_l} p_M(\boldsymbol{\theta})$:

$$\partial_{\theta_l} p_M(\boldsymbol{\theta}) = \text{Tr}\{\partial_{\theta_l} \tilde{\rho_M}(\boldsymbol{\theta})\} \tag{C.27}$$

$$= \langle 0 | \left( \overrightarrow{\prod_{m=1}^{M-1}} U_m^\dagger(\theta_m)\Pi_m \right) U_M^\dagger(\theta_M)\Pi_M U_M(\theta_M) \left( \overleftarrow{\prod_{m=l+1}^{M-1}} \Pi_m U_m(\theta_m) \right) \Pi_l \partial_{\theta_l} U_l(\theta_l) \tag{C.28}$$

$$\times \left( \overleftarrow{\prod_{m=1}^{l-1}} \Pi_m U_m(\theta_m) \right) |0\rangle + \langle 0 | \left( \overrightarrow{\prod_{m=1}^{l-1}} U_m^\dagger(\theta_m)\Pi_m \right) \partial_{\theta_l} U_l^\dagger(\theta_l)\Pi_l \left( \overrightarrow{\prod_{m=l+1}^{M-1}} U_m^\dagger(\theta_m)\Pi_m \right) \tag{C.29}$$

$$\times U_M^\dagger(\theta_M)\Pi_M U_M(\theta_M) \left( \overleftarrow{\prod_{m=1}^{M-1}} \Pi_m U_m(\theta_m) \right) |0\rangle \tag{C.30}$$

$$= \langle \tilde{\psi}_0 | U_l^\dagger(\theta_l) \tilde{\Pi}_M \partial_{\theta_l} U_l(\theta_l) | \tilde{\psi}_0 \rangle + \langle \tilde{\psi}_0 | \partial_{\theta_l} U_l^\dagger(\theta_l) \tilde{\Pi}_M U_l(\theta_l) | \tilde{\psi}_0 \rangle , \tag{C.31}$$

where

$$\tilde{\Pi}_M = \left( \overrightarrow{\prod_{m=l+1}^{M-1}} U_m^\dagger(\theta_m)\Pi_m \right) U_M^\dagger(\theta_M)\Pi_M U_M(\theta_M) \left( \overleftarrow{\prod_{m=l+1}^{M-1}} \Pi_m U_m(\theta_m) \right), \tag{C.32}$$

and $|\tilde{\psi}_0\rangle$ is the same as in Eq. (C.17). Again, we can apply the parameter-shift rule to obtain

$$\partial_{\theta_l} p_M(\boldsymbol{\theta}) = \frac{1}{2} \left( \langle \tilde{\psi}_0 | U^\dagger\left(\theta_l + \frac{\pi}{2}\right) \tilde{\Pi}_M U\left(\theta_l + \frac{\pi}{2}\right) - U^\dagger\left(\theta_l - \frac{\pi}{2}\right) \tilde{\Pi}_M U\left(\theta_l - \frac{\pi}{2}\right) | \tilde{\psi}_0 \rangle \right). \tag{C.33}$$

But these expectation values are simply the normalization constants $p_M^{\pm,l}$ of Eq. (C.22), hence the final result becomes

$$\text{Tr}\{\tilde{\rho}_M(\boldsymbol{\theta})(\partial_{\theta_l} p_M^{-1}(\boldsymbol{\theta}))H\} = -\langle H \rangle_{\boldsymbol{\theta}}\, \frac{1}{2} \left( \frac{p_M^{+,l}}{p_M} - \frac{p_M^{-,l}}{p_M} \right). \tag{C.34}$$

Combining (i) and (ii) we finally obtain the projective gradient:

$$\partial_{\theta_l} \langle H_M \rangle_{\boldsymbol{\theta}} = \frac{1}{2} \left( \left( \langle H \rangle_{\boldsymbol{\theta}}^+ - \langle H \rangle_{\boldsymbol{\theta}} \right) \frac{p_M^{+,l}}{p_M} - \left( \langle H \rangle_{\boldsymbol{\theta}}^- - \langle H \rangle_{\boldsymbol{\theta}} \right) \frac{p_M^{-,l}}{p_M} \right). \tag{C.35}$$

To calculate these projective gradients and produce Fig. 4 in the main text, we use the TensorFlow-based quantum simulator Zyglrox [63]. Note that in practice, estimating these gradients will be exponentially difficult due to the ratio $p_M^{-,l}/p_M$.

# D  A practical optimization algorithm with projective measurements

Any state $\rho_M(\boldsymbol{\theta})$ is weighted by two probabilities: a classical and quantum probability. The former is the result of flipping a coin with probability $p$ after each layer for each qubit, which results in a measurement configuration. The latter is the quantum probability (obtained via the Born rule) of measuring an outcome of the particular measurement configuration, which we've denoted by $p_M(\boldsymbol{\theta})$.

We can denote the classical probability of a measurement configuration in layer $m$ with

$$s_m^{(c_m)} = \prod_{j=1}^{N} p^{\mathbb{I}(c_{m,j}=0)} (1-p)^{\mathbb{I}(c_{m,j}=1)}, \tag{D.1}$$

where $c_{m,j} = 0$ indicates that we perform a measurement and $c_{m,j} = 1$ indicates that we do not. The tuple $c_m = (c_{m,0}, \ldots, c_{m,N})$ thus labels the a measurement setting in $m$. The total probability over all layers is then given by the product of these individual layer-wise probabilities:

$$s(c) = \prod_{m=1}^{M} s_m^{(c_m)}. \tag{D.2}$$

The tuple $c = (c_1, \ldots, c_M)$ then labels a possible measurement setting.

After choosing a measurement setting, we run the circuit and perform the measurements. This results in a set of outcomes $i = (i_1, \ldots, i_M)$, where $i_m = (i_{m,1}, \ldots, i_{m,N})$ indicates the outcomes per layer. The integer $i_{m,j} \in \{0, 1, 2\}$ with $j = 1, \ldots, N$ indicates the measurement of $|0\rangle\langle 0|$, $|1\rangle\langle 1|$ and the identity operator, respectively. We now explicitly denote with $\rho_M(i, c, \boldsymbol{\theta})$ the state resulting from a particular measurement setting, and with $p_M(i|\boldsymbol{\theta}; c)$ the probability of obtaining a particular outcome $i$, given a measurement setting $c$.

If we remix the resulting pure states $\rho_M(i, c, \boldsymbol{\theta})$ according to the classical probability $s^{(i)}$ and quantum probability $p_M^{(i)}$ from this into a single density matrix, we obtain

$$\varrho = \sum_{i,c} s(c) p_M(i|\boldsymbol{\theta}; c)(\boldsymbol{\theta}) \rho_M(i, c, \boldsymbol{\theta}). \tag{D.3}$$

---

**Algorithm D1:** Algorithm to obtain the gradient of Eq. (D.7)

---

**Input:** $\varrho_0$, $U(\boldsymbol{\theta})$, $H$, $p$, $N_s$, $\theta_l$
$h^{+,l} \leftarrow 0$
$h^{-,l} \leftarrow 0$
**for** $n \in (1, \ldots, N_s)$ **do**
    Create measurement configuration **for** $m \in (1, \ldots, m)$ **do**
        **for** $j \in (1, \ldots, N)$ **do**
            $c_{m,j} \sim \mathrm{Ber}(p)$
    $\theta_l \leftarrow \theta_l + \pi/2$
    Run $U(\boldsymbol{\theta})$ with measurement setting $c$, obtain outcomes $i$ and state $\rho_M^{+,l}(i|\boldsymbol{\theta}; c)$.
    Measure $H$ and obtain eigenvalue $h$ $h^{+,l} \leftarrow h^{+,l} + h$
    $\theta_l \leftarrow \theta_l - \pi$
    Run $U(\boldsymbol{\theta})$ with measurement setting $c$, obtain outcomes $i'$ and state $\rho_M^{-,l}(i'|\boldsymbol{\theta}; c)$.
    Measure $H$ and obtain eigenvalue $h'$
    $h^{-,l} \leftarrow h^{-,l} + h'$
**Output:** $\frac{1}{2}(h^{+,l} - h^{-,l})$

---

We can calculate a variational energy with respect to this density matrix as

$$E_{\text{int}}(\boldsymbol{\theta}) = \sum_{i,c} s(c) p_M(i|\boldsymbol{\theta};c)(\boldsymbol{\theta}) \operatorname{Tr}\{\rho_M(i,c,\boldsymbol{\theta})H\}, \tag{D.4}$$

where $H$ is a Hermitian operator. Clearly $E_{\text{ground}} \leq E_{\text{int}}(\boldsymbol{\theta})$. Calculating the gradient of Eq. (D.4) involves calculating the gradient for all individual states in the mixture. Note that the mixture in Eq. (D.4) can be written as a sum of unnormalized states

$$E_{\text{int}}(\boldsymbol{\theta}) = \sum_{i,c} s(c) p_M(i|\boldsymbol{\theta};c)(\boldsymbol{\theta}) \operatorname{Tr}\{\tilde{\rho}_M(i,c,\boldsymbol{\theta}) p_M^{-1}(i|\boldsymbol{\theta};c)H\} \tag{D.5}$$

$$= \sum_{i,c} s(c) \operatorname{Tr}\{\tilde{\rho}(i,c,\boldsymbol{\theta})H\}. \tag{D.6}$$

From Eq. (C.25) we then see immediately that the gradient of the mixed state is then given by

$$\operatorname{Tr}\{(\partial_{\theta_l}\rho)H\} = \sum_{i,c} \frac{s(c)}{2} \left( \langle H \rangle_{\boldsymbol{\theta},c,i}^{+,l} p_M^{+,l}(i|\boldsymbol{\theta};c) - \langle H \rangle_{\boldsymbol{\theta},c,i}^{-,l} p_M^{-,l}(i|\boldsymbol{\theta};c) \right). \tag{D.7}$$

Hence the estimator for the gradient corresponds to the average expectation value over intermediate measurements done on parameter-shifted circuits weighted by $p_M^{(i),l}$ and the classical probability $s^{(i)}$. Therefore, the projective gradients can be estimated by obtaining statistics from the measurements done on the parameter-shifted circuits. Given a number of shots $N_s$, the gradient of Eq. (D.7) can be obtained with Alg. D1.

# E   Data collapse of the projective gradients

To observe the phase transition in the variance of the projective gradients of Fig. E2, we perform a data collapse of the following quantity:

$$\log\left(\text{Var}\left[\left.\frac{\partial ZZ}{\partial \theta_0}\right]\right|_{N,p}\right) = \log\left(\text{Var}\left[\left.\frac{\partial ZZ}{\partial \theta_0}\right]\right|_{N,p_c}\right) + g\left(N^{1/\nu}(p-p_c)\right). \tag{E.1}$$

We use the same method as in App. A. The resulting data collapse can be seen in Fig. 4.

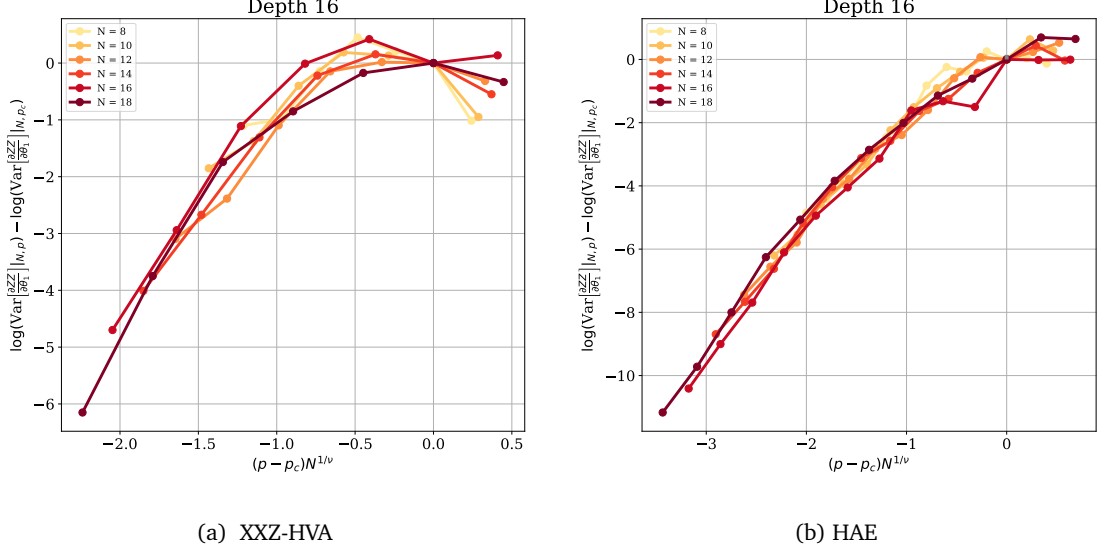

(a) XXZ-HVA

(b) HAE

Figure E2: Data collapse for the projective gradients at $p_c = 0.25$ and $p_c = 0.5$ for the XXZ-HVA and HAE circuits, respectively. Since the data shown in Fig. 4 is noisy, the data collapse is not as clean, especially for the XXZ-HVA circuit. However, we still find critical exponents that are close to the ones obtained from the entanglement entropy scaling collapse, with $\nu \approx 1.31$ and $\nu \approx 1.5$, respectively.

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
