# Peer review of "Measurement-induced entanglement phase transitions in variational quantum circuits"

_SciPost Physics, doi:SciPost Phys. 14, 147 (2023)_

## Round 1 · Referee Report · Anonymous (Referee 1) · 2022-6-29

Strengths

  • original idea
  • well motivated
  • can stimulate novel work

Weaknesses

  • cryptic presentation
  • a bit of overselling
  • no effort in bridging between readerships

Report

The work by Wiersema et al. attempts a connection between MIPT and variational algorithms. Their research appears well justified and poses a number of stimulating questions for the future of the field of digital quantum simulators. However, I cannot recommend the paper for publication in the current form for a number of presentation issues, that are borderline with hindering understanding, as I detail in the following. In order to provide constructive criticism, I should say that most of the issues would be probably solved by formatting the paper in a style more SciPost-friendly. This appears to me a Letter, as the authors themselves write at page 1 of the manuscript.

To start with, the introduction could be improved to a large extent. There are a number of sentences which appear too colloquial. Examples range from poor effort in connecting blocks: ‘Another important element ..’, ‘however, this is not the case’,
to statements that do not appear solid: what does it mean ‘entanglement is destroyed globally’? to the motivation given at page 1: ‘also take place in circuits of practical interest…machine learning’. The work done by the authors is to look at quantities interesting for variational algorithms, not really changing approach by looking at new circuits, as I discuss in the following. All of this may appear cosmetics but it accumulates through the manuscript and it leaves a bad feeling to the reader.

The motivation given at page 2 (first column) is instead way better summarized, yet the authors should have now the space to elaborate a bit more on barren plateaux, connecting the reader not familiar with the topic. It is hard for me to follow that part, besides getting a sense of its relevance.

From Fig 2 it appears that the authors are not solving for Floquet dynamics of an interacting integrable model interspersed with measurements (what they claim is an open question). Instead, they alternate XX, YY, XX gates. This seems very similar in spirit to what several other authors have done with Clifford circuits et similia, and indeed the authors do find analogue results.
Again, this would have been a less critical report if the related statement at the beginning of page 3 had been milder.

Finally, I appreciate the results in Fig 3, but they are presented too fast and way too cryptical. Which observable is O? why parameters thata_l are shifted? Also, am I lost or Eq 4 is linear in the density matrix? if so, how it can lnow about the MIPT? this is the whole point about such field. Perhaps, I am just confused by the too compact presentation, but these aspects should be definitely amended in the next version of the paper. In the same block, there is a typo at page 4: Fig 2 does not discuss circuit settings, but data collapse for entanglement entropies. Another hint that perhaps the authors should invest more time in the presentation of their results.

Requested changes

  • enlarge the manuscript
  • downplay some statements
  • expand significantly the original part of the manuscript (currently page 4)
  • address my perplexities on Eq 4

  • validity: ok
  • significance: good
  • originality: good
  • clarity: ok
  • formatting: acceptable
  • grammar: good

Author:  Roeland Wiersema  on 2022-12-07  [id 3110]

(in reply to Report 1 on 2022-06-29)

General Comments

1.) To start with, the introduction could be improved to a large extent. There are a number of sentences which appear too colloquial. Examples range from poor effort in connecting blocks: ‘Another important element ..’, ‘however, this is not the case’, to statements that do not appear solid: what does it mean ‘entanglement is destroyed globally’? to the motivation given at page 1: ‘also take place in circuits of practical interest. . . machine learning’.

Authors: We significantly revised the introduction following the referees’ comments. We have removed colloquial sentences and improved overall flow of the text. In addition, we now try to emphasize the main message of our work in the introduction: connecting two active areas of research in condensed matter and variational quantum computing.

2.) The work done by the authors is to look at quantities interesting for variational algorithms, not really changing approach by looking at new circuits, as I discuss in the following. All of this may appear cosmetics but it accumulates through the manuscript and it leaves a bad feeling to the reader.

Authors: As the referee points out, we are indeed not investigating new circuits. Instead, we are looking at the effect of measurements in known variational quantum circuits used to solve a variety of different tasks and we take the ground state estimation problem as a prototypical example of a VQA. We confirm that measurement induced entanglement phase transitions take place, and show that these transitions coincide with a transition in the gradient variance of local observables. We outline the possibility to use this effect to improve the trainability of variational quantum circuits and we made an effort to more clearly emphasize these points in the main text.

Specific Comments

1.) The motivation given at page 2 (first column) is instead way better summarized, yet the authors should have now the space to elaborate a bit more on barren plateaux, connecting the reader not familiar with the topic. It is hard for me to follow that part, besides getting a sense of its relevance.

Authors: We have added the following explanation of barren plateaus at the bottom of page 2: “Secondly, it is known that randomly initialized variational quantum circuits tend to approximate unitary 2-designs, which are known to have exponentially decaying gradients with respect to the gate parameters as a function of system size. These so-called barren plateaus pose a significant hurdle for variational quantum algorithms, since the number of measurements required to accurately estimate the gradients quickly becomes intractable [1, 2, 3]"

2.) From Fig 2 it appears that the authors are not solving for Floquet dynamics of an interacting integrable model interspersed with measurements (what they claim is an open question). Instead, they alternate XX, YY, XX gates. This seems very similar in spirit to what several other authors have done with Clifford circuits et similia, and indeed the authors do find analogue results. Again, this would have been a less critical report if the related statement at the beginning of page 3 had been milder.

Authors: We indeed investigate circuits that are similar to the random circuits studied in previous works, with the main difference being that our circuits are of specific interest to the variational quantum computing community. As the referee rightfully points out, there are many works on random unitary circuits with non-integrable dynamics that lead to MIPT. Although we do not evolve the state exactly according to the Heisenberg dynamics, our circuit dynamics can be seen as applying random quenches to the state under the XXZ Hamiltonian, which can be studied analytically via the Bethe ansatz [4, 5]. We have added the following clarification in the main text: “Here, we address a closely related model, where the unitary dynamics are generated by random quenches under the XXZ Hamiltonian."

Additionally, our circuits are not providing Floquet dynamics, since they are non-deterministic: we randomly sample the parameters of the circuit. This is mentioned below figure 3. in the revised text: “The average $S(p, N )$ is obtained by averaging over $3 × 10^3$ circuit realizations with all circuit parameters sampled uniformly in $(0, 2π)$"

3.) Finally, I appreciate the results in Fig 3, but they are presented too fast and way too cryptical. Which observable is O? why parameters θl are shifted? Also, am I lost or Eq 4 is linear in the density matrix? If so, how it can know about the MIPT? this is the whole point about such field. Perhaps, I am just confused by the too compact presentation, but these aspects should be definitely amended in the next version of the paper. In the same block, there is a typo at page 4: Fig 2 does not discuss circuit settings, but data collapse for entanglement entropies. Another hint that perhaps the authors should invest more time in the presentation of their results.

Authors: We have made an effort to explain the results in more detail. The section on the numerical results for projective gradients and the barren plateau now starts with a recap of the parameter-shift rule. We also ensure that the setup of the numerical experiments is presented more clearly and we reference the correct circuit settings.

With regards to the linearity of the gradients with the density matrix: yes, the cost function is linear in the density matrix, but it is a highly nonlinear function of the circuit parameters. The reason why the gradients may feel the effect of the MIPT is because there seems to be an intricate connection between gradient magnitude and the entanglement in the circuit. For instance in [ 6], the amount of entanglement in the circuit can be used to bound the gradient variance. In [7], it is heuristically shown that limiting entanglement improves the quality of the gradients in early circuit optimization. Finally, in [ 8 ] it is shown that circuits that are far away from 2-designs, which are known to follow volume law entanglement scaling are more easy to optimize. To summarize, it seems that the entanglement in the circuit seems to be intricately linked to the gradients in a circuit. Since the MIPT affects the entanglement in the circuit, it therefore also affects the gradients.

References [1] Jarrod R. McClean, Sergio Boixo, Vadim N. Smelyanskiy, Ryan Babbush, and Hartmut Neven. Barren plateaus in quantum neural network training landscapes. Nature Communications, 9(1):4812, Nov 2018. [2] M. Cerezo, Akira Sone, Tyler Volkoff, Lukasz Cincio, and Patrick J. Coles. Cost function dependent barren plateaus in shallow parametrized quantum circuits. Nature Communications, 12(1):1791, Mar 2021. [3] Samson Wang, Enrico Fontana, M. Cerezo, Kunal Sharma, Akira Sone, Lukasz Cincio, and Patrick J. Coles. Noise-Induced Barren Plateaus in Variational Quantum Algorithms, 2021. arXiv:2007.14384. [4] Vincenzo Alba and Pasquale Calabrese. Entanglement and thermodynamics after a quantum quench in integrable systems. Proceedings of the National Academy of Sciences, 114(30):7947–7951, 2017. [5] Adam Nahum, Jonathan Ruhman, Sagar Vijay, and Jeongwan Haah. Quantum Entanglement Growth under Random Unitary Dynamics. Phys. Rev. X, 7:031016, Jul 2017. [6] Carlos Ortiz Marrero, Mária Kieferová, and Nathan Wiebe. Entanglement Induced Barren Plateaus, 2020. arXiv:2010.15968. [7] Taylor L. Patti, Khadijeh Najafi, Xun Gao, and Susanne F. Yelin. Entanglement Devised Barren Plateau Mitigation, 2020. arXiv:2012.12658. [8] Zoë Holmes, Kunal Sharma, M. Cerezo, and Patrick J. Coles. Connecting ansatz expressibility to gradient magnitudes and barren plateaus. PRX Quantum, 3:010313, Jan 2022.

---

## Round 1 · Referee Report · Anonymous (Referee 2) · 2022-7-28

Report

The aim of this paper is to make a connection between measurement-induced entanglement phase transitions in random circuits and variational quantum circuits proposed for quantum machine learning. The authors investigate the possibility of using measurements, which reduce entanglement in the circuits, to avoid the barren plateau problem. They propose numerical evidence for such a connection. This is a very interesting direction and the paper is generally well-written and interesting.

However, I would also like to see a clarification of the issue pointed out by the other referee, namely Equation 4 and the physical meaning of the derivatives being calculated.

The authors discuss the dependence of the output mixed state upon parameters theta. However, this mixed state, which is averaged over measurement outcomes, is well known to be insensitive to the measurement-induced entanglement transition.

After Equation 4 the authors write rho(theta) as a function of p^i and rho^i(theta). But p^i is also a function of theta. Is p^i also differentiated in Equation 4? If not, what is the physical meaning of this derivative?

Other specific comments:

Although the circuits discussed here are termed “variational”, the parameters are not optimized. Instead they are sampled uniformly, giving a random circuit ensemble, which the authors envisage as a starting point for a variational calculation. This is a potential source of confusion for the reader, so it would be useful to have a clarifying comment early on.

Page 2-3 It is stated that the HVA circuit is integrable: clarify exactly what is integrable (e.g. is it the unitary circuit for arbitrary parameter values but without measurements? or the unitary circuit with particular parameter values?)

Page 2 mapping to 2 dimensional percolation was described in ref 5 (this ref also relevant to “steady state” entanglement just below).

Page 3 it is stated that measurements are sampled uniformly. Does this mean that the standard quantum mechanical measurement probability is not used?

One of the main claims of the paper is that Fig 3 shows a transition that coincides with the measurement induced entanglement transition. However, an independent determination of a transition point (or bounds on such a point) from the data in Fig 3, is lacking.

More general comments:

Do the authors see applications of the phase transition in ensembles where the parameters are in fact variationally optimized?

The authors propose using these measurement circuits as a practical tool for variational optimization of a wavefunction. However the measurement makes the output wavefunction stochastic, as they discuss. This stochasticity may be harmful for targeting a particular state. Therefore it may be useful to also include feedback based on the measurement outcomes. E.g. it is known that some topologically ordered states can be prepared deterministically by shallow depth circuits if measurement and feedback is allowed.
  • validity: -
  • significance: -
  • originality: -
  • clarity: -
  • formatting: -
  • grammar: -

Author:  Roeland Wiersema  on 2022-12-07  [id 3109]

(in reply to Report 2 on 2022-07-28)

General Comments

1.) Do the authors see applications of the phase transition in ensembles where the parameters are in fact variationally optimized?

Authors: Yes, it seems that the source of training difficulty in variational circuits comes from initializing a circuit randomly with a high entanglement state. The measurements we introduce are key for creating low entanglement states for which barren plateaus may be ameliorated. The problem is how one can calculate gradients for these ensembles to minimize the cost function, since the gradients we study might be hard to calculate in practice. We provide one possible method for calculating these gradients and outline what a variational algorithm for ground state optimization could look like in Appendix D. However, there will be practical difficulties with these algorithms that we hope to explore in the future.

2.) The authors propose using these measurement circuits as a practical tool for variational optimization of a wavefunction. However the measurement makes the output wavefunction stochastic, as they discuss. This stochasticity may be harmful for targeting a particular state. Therefore it may be useful to also include feedback based on the measurement outcomes. E.g. it is known that some topologically ordered states can be prepared deterministically by shallow depth circuits if measurement and feedback is allowed.

Authors: This is a very interesting suggestion that we have not investigated in detail. How one would make direct use of the measurement outcomes in a gradient-based optimization is not obvious to us right now. The scheme we envisioned and have experimented with is annealing the measurement rate during variational optimization to obtain a pure state at end of training. Hence we would have a stochastic algorithm at the start of the optimization but a deterministic (up to gradient estimation errors) at the end of the optimization.

Specific Comments

1.)The authors discuss the dependence of the output mixed state upon parameters theta. However, this mixed state, which is averaged over measurement outcomes, is well known to be insensitive to the measurement- induced entanglement transition.

Authors: We agree that the mixed state itself is insensitive to the measurement induced phase transition. The full mixture of the individual pure states obtained after the circuit with intermediate measurements will tend to a system at infinite temperature. As a result, the gradients with respect to this state will likely inhibit barren plateaus again [1]. However, gradients with respect to the pure states seem to not suffer from this, which is our main result. The former gradients can be estimated with little trouble, whereas the latter gradients are exponentially hard to obtain. We have added clarifications in the text to separate these two more clearly and emphasize the latter result: individual worldline gradients do not suffer from barren plateaus.

2.) I would also like to see a clarification of the issue pointed out by the other referee, namely Equation 4 and the physical meaning of the derivatives being calculated. After Equation 4 the authors write $ρ(θ)$ as a function of $p_i$ and $ρ_i(θ)$. But $p_i$ is also a function of $θ$. Is $p_i$ also differentiated in Equation 4? If not, what is the physical meaning of this derivative?

Authors: We have made it clearer what is being calculated in Figure 4: these are gradients with respect to the individual pure states obtained from the circuit interspersed with measurements. Due to the difficulty to obtain these gradients in practice, we suggest measuring gradients with respect to the full mixture. However, as noted in the point above, the remixing of the different worldlines may again result in a barren plateau.

3.) Although the circuits discussed here are termed “variational”, the parameters are not optimized. Instead they are sampled uniformly, giving a random circuit ensemble, which the authors envisage as a starting point for a variational calculation. This is a potential source of confusion for the reader, so it would be useful to have a clarifying comment early on.

Authors: We have added a clarification for this point in Section 3 below equation (5):

4.) “Also, we are not performing any optimization; we consider the variational circuit at initialization." 2.4 Page 2-3 It is stated that the HVA circuit is integrable: clarify exactly what is integrable (e.g. is it the unitary circuit for arbitrary parameter values but without measurements? or the unitary circuit with particular parameter values?)

Authors: By integrable, we mean that there exists an exact solution for the energy spectrum of the Hamiltonian that generates the dynamics of out circuit. We have added the following clarification to the main text above figure 2:

“The HVA for the XXZ model is of particular interest since the XXZ Hamiltonian is Bethe-ansatz integrable, i.e. there exists an analytical solution for the energy spectrum. Additionally, the entan- glement properties of these systems undergoing quenches can be understood analytically [2 , 3]. For such models, it is still an open question if the corresponding unitary dynamics interspersed with measurements will produce a measurement-induced entanglement phase transitions [ 4]. Here, we address a closely related model, where the unitary dynamics are generated by random quenches under the XXZ Hamiltonian."

5.) Page 2 mapping to 2 dimensional percolation was described in ref 5 (this ref also relevant to “steady state” entanglement just below).

Authors: We have added this reference at these locations.

6.) Page 3 it is stated that measurements are sampled uniformly. Does this mean that the standard quantum mechanical measurement probability is not used?

Authors: We mean that the measurements after each layer for each qubit are performed with probability $p$, but the applied projector is determined via the standard mechanical measurement probability $p = \mathrm{Tr}{ρΠ}$ where $Π$ is a projector onto an eigenstate of the observable $O$. We have clarified this in the text below equation (4) with the following sentence: “Which projector $Π_i$ is applied depends on the quantum probability $ \mathrm{Tr}{Π_iρ}$. "

7.) One of the main claims of the paper is that Fig 3 shows a transition that coincides with the measurement induced entanglement transition. However, an independent determination of a transition point (or bounds on such a point) from the data in Fig 3, is lacking.

Authors: We have added the data for this in Appendix E.

References [1] Samson Wang, Enrico Fontana, M. Cerezo, Kunal Sharma, Akira Sone, Lukasz Cincio, and Patrick J. Coles. Noise-Induced Barren Plateaus in Variational Quantum Algorithms, 2021. arXiv:2007.14384. [2] Vincenzo Alba and Pasquale Calabrese. Entanglement and thermodynamics after a quantum quench in integrable systems. Proceedings of the National Academy of Sciences, 114(30):7947–7951, 2017. [3] Adam Nahum, Jonathan Ruhman, Sagar Vijay, and Jeongwan Haah. Quantum Entanglement Growth under Random Unitary Dynamics. Phys. Rev. X, 7:031016, Jul 2017. [4] Yimu Bao, Soonwon Choi, and Ehud Altman. Theory of the phase transition in random unitary circuits with measurements. Phys. Rev. B, 101:104301, 3 2020.

---

## Round 2 · Referee Report · Anonymous (Referee 1) · 2022-12-12

Strengths

1- clear goal 2- well presented 3- intriguing bridge between two communities

Weaknesses

1- unclear whether the findings of the paper can really have useful impact on q. algorithms

Report

The authors have seriously considered my remarks, rewriting large part of the paper. The manuscript does indeed address a topic suited to SciPost standards and I am happy to recommend the paper for acceptance.

I have one last important physical question: the volume law barrier the authors talk about may have in my opinion little to do with MIPTs. The latter relies on post-selecting trajectories, and without this procedure, the asymptotic state is an infinite temperature one, with boring entanglement.
Otherwise, if MIPTs could be relevant for their variational algorithms, then we would have found a remarkable application of measurement induced transitions, which are very elusive as the authors know.

Could they comment on this issue? I think it is important since it seems to undermine a bit their original motivation
  • validity: -
  • significance: -
  • originality: -
  • clarity: -
  • formatting: -
  • grammar: -

Author:  Roeland Wiersema  on 2022-12-21  [id 3172]

(in reply to Report 1 on 2022-12-12)

We agree that the MIPT relies on the post-selected trajectories which are hard to access in a variational algorithm. We concede this point just above Section 5 after exploring what a variational algorithm that includes remixed trajectories could look like in appendix D. As mentioned by the referee, such an algorithm would effectively be starting with with an infinite temperature state, with rather boring entanglement that would make optimization difficult. Hence although the original motivation for this work was to indeed find a way to utilize the MIPT in a variational algorithm, we found that this may be difficult in practice.

However, we believe that our numerical results for the barren plateau effect with respect to the individual trajectories of the measured circuits support the idea that the trainability of variational algorithms is intricately linked to the amount of entanglement produced in the circuit. We therefore hope that our work serves as further motivation to investigate this connection in more detail.

---

## Round 2 · Referee Report · Anonymous (Referee 2) · 2022-12-15

Report

The authors appear to have made a serious attempt to address the comments of the referees and to take better account of the challenges in relating the transition they discuss to practical algorithms.

---

## Round 2 · Author Response

We want to thank the referees for their careful review and the suggestions for improving the manuscript. In
addition to addressing their comments we have rewritten a large portion of the original paper.

---

## Round 2 · List of Changes

1. Restructured the text, expanded some of the sections and adapted the formatting to be suitable for a SciPost article.
  2. Added Appendix D where we propose a possible practical algorithm that uses intermediate measurement in an optimization setting.
  3. Added Appendix E where we perform a finite-scaling analysis of the critical exponents through the gradient variance.
  4. Added Figure 1 to illustrate the general measurement setup we consider to support the mathematical exposition in the main text and appendix.

---

## Round 3 · List of Changes

- No changes

---

## Editorial Decision

published